# Thermal Conductivity of Nano-Crystallized Indium-Gallium-Zinc Oxide Thin Films Determined by Differential Three-Omega Method

**DOI:** 10.3390/nano11061547

**Published:** 2021-06-11

**Authors:** Rauf Khan, Michitaka Ohtaki, Satoshi Hata, Koji Miyazaki, Reiji Hattori

**Affiliations:** 1Department of Applied Science for Electronics and Materials, Interdisciplinary Graduate School of Engineering Sciences, Kyushu University, Fukuoka 816-8580, Japan; ohtaki@kyudai.jp (M.O.); hata.satoshi.207@m.kyushu-u.ac.jp (S.H.); hattori@gic.kyushu-u.ac.jp (R.H.); 2Transdisciplinary Research and Education Center for Green Technologies, Interdisciplinary Graduate School of Engineering Sciences, Kyushu University, Fukuoka 816-8580, Japan; 3The Ultramicroscopy Research Centre, Kyushu University, Fukuoka 819-0395, Japan; 4Department of Mechanical and Control Engineering, Kyushu Institute of Technology, Fukuoka 804-8550, Japan; miyazaki@mech.kyutech.ac.jp

**Keywords:** IGZO thin film, three-omega method, thermal conductivity, nano crystallinity, transmission electron microscopy

## Abstract

The temperature dependence thermal conductivity of the indium-gallium-zinc oxide (IGZO) thin films was investigated with the differential three-omega method for the clear demonstration of nanocrystallinity. The thin films were deposited on an alumina (α-Al_2_O_3_) substrate by direct current (DC) magnetron sputtering at different oxygen partial pressures ([PO_2_] = 0%, 10%, and 65%). Their thermal conductivities at room temperature were measured to be 1.65, 1.76, and 2.58 Wm^−1^K^−1^, respectively. The thermal conductivities decreased with an increase in the ambient measurement temperature. This thermal property is similar to that of crystalline materials. Electron microscopy observations revealed the presence of nanocrystals embedded in the amorphous matrix of the IGZO films. The typical size of the nanocrystals was approximately 2–5 nm with the lattice distance of about 0.24–0.26 nm. These experimental results indicate that the nanocrystalline microstructure controls the heat conduction in the IGZO films.

## 1. Introduction

Indium-gallium-zinc oxide (IGZO) thin-film transistors (TFTs) have attracted considerable research attention in the area of TFT technology. Their electronic device applications are being enhanced owing to their remarkable performance, e.g., high electron carrier mobility, better uniformity over a large area, and potential optical transparency [1,2,3,4]. Compared with low-temperature polysilicon (LTPS) and hydrogenated amorphous silicon (a-Si:H) TFTs, the IGZO TFT technologies are rapidly gaining popularity for enabling flexible or stretchable displays [5,6,7,8]. Moreover, ZnO active-channel TFTs are promising solutions to high-resolution display technologies but are not suitable for large-area display applications owing to their small feature sizes [9]. Hsieh et al. fabricated ZnO TFTs with varying active-channel lengths and widths and achieved a saturation region mobility of >8 cm^2^/Vs [10]. Concurrently, Yabuta et al. employed amorphous IGZO (a-IGZO) active channels in TFTs and observed good electrical properties, which indicate the possibility of large-area, high-resolution, and integrated circuit (IC) applications [2]. In transparent IGZO TFTs, it was first used as a single crystalline active channel material because of its high electron mobility (80 cm^2^/(Vs)) [11]. On the other hand, a-IGZO is also popular for stretchable or flexible electronics for its capability to be fabricated at room temperature with high mobility (µ > 10 cm^2^/(Vs)) [12]. Moreover, IGZO can be a promising thermoelectric candidate for its comparable thermoelectric performance to other oxide candidates [13]. The thermal conductivity of IGZO indicates its ability for heat conduction, which is subject to various factors, and can be subdivided into electro- and phono-thermal conductivities revealed by phonon–phonon and electron–electron scattering [14]. Moreover, IGZO is also used in the photovoltaic systems due to its large-area competence [10,15]. It is also expected to use as a flexible solar cell for its low processing temperature [16]. Most studies on IGZO thin films focused on electrical, electronic, and optical properties [17]. The thermal properties of IGZO thin films are also of vital importance for heat dissipation and thermal management of devices [18,19,20]. Although the atomic-scale structure of IGZO films has been investigated to reveal whether these are amorphous or crystalline [21,22,23], few fundamental studies have been performed on the relationship between the thermal conductivity and microstructure of IGZO thin films [24].

In this study, the temperature dependence of the thermal conductivity of IGZO thin films was measured using the differential three-omega (3ω) method. The thin films with various O_2_/Ar gas flow ratios were deposited on an alumina substrate by the DC magnetron sputtering method. The result of thermal conductivity reveals a clear conclusion on the crystalline phase. The oxygen partial pressure [PO_2_] of deposited films are 0%, 10%, and 65%. This is because the gas flow ratios have inversely proportional relationships with carrier density and electrical conductivity. The 3ω method is highly accurate and widely implemented for cross-sectional plane thermal conductivity measurements. In particular, it is utilized to investigate the thermal properties of thin films [25,26]. Scanning transmission electron microscopy (STEM) observations were also carried out to characterize the nanometer-scale microstructure of the IGZO thin film.

## 2. Materials and Methods

### 2.1. Thin Film Preparation

A 500 nm IGZO (In:Ga:Zn = 1:1:1) thin films with were deposited on an alumina substrate by the DC magnetron sputtering method at room temperature. The deposition rate of this method is higher than others such as pulsed laser deposition (PLD), radio frequency (RF) sputtering [27]. A mixture of Ar and O_2_ gas was used during the sputtering process. The oxygen partial pressure [PO_2_] was calculated from the ratio of O_2_ gas flow rate and total Ar/O_2_ flow rates. Three samples at various [PO_2_] such as [PO_2_] = 0% (Sample A), [PO_2_] = 10% (Sample B, and [PO_2_] = 65% (Sample C) were deposited for pursuing the measurement. Figure 1 shows a plan view of the dimension of shadow mask and a cross-sectional view of the film samples and reference. An aluminum wire strip is deposited on the IGZO thin film in vacuum using 30 µm width and 2 mm length shadow mask for three-omega measurement [28].

### 2.2. Transmission Electron Microscopy Observation

We performed focused Ga-ion beam milling in a dual-beam apparatus (Thermo Fisher Scientific Versa 3D, Agilent Instrument, Santa Clara, CA, USA) to prepare the cross-sectional sample. Before performing focused ion beam (FIB) milling, we deposited platinum (Pt), tungsten (W), and carbon (C) protection layers on the IGZO thin film. First, in the FIB milling process, we milled both sides of the cross-sectional sample of the IGZO thin film down to a thickness of 1.0 µm at 30 kV and 1.0 nA. Second, this sample was thinned down to 0.5 µm at 30 kV and 0.5 nA. Third, the sample was milled down further to 0.3 µm at 30 kV and 0.3 nA. Finally, we performed FIB milling at 30 kV and 0.1 nA to obtain a cross-sectional sample with a thickness less than 50 nm. Figure 2 shows the cross-section of secondary-ion microscopy of the FIB specimen. After the FIB milling processes, additional Ar ion milling (Gatan PIPS II, Gatan Inc, Pleasanton, CA, USA and Fischione Nano Mill, Fischione Instrument, Export, PA, USA) was performed to remove the damaged layer formed on the sample surfaces by the Ga ion bombardment (Figure 2b). Transmission electron microscopy (TEM) and scanning transmission electron microscopy (STEM) observations were performed using a Thermo Fisher Scientific Titan G2 60-300 (Thermo Fisher Scientific, Waltham, MA, USA) and JEOL ARM-200F (JEOL, Tokyo, Japan) electron microscope equipped with spherical aberration correctors.

### 2.3. Three-Omega Method

The 3ω method is a highly simple, accurate and effective thermal conductivity measuring technique of thin films and bulk materials [29,30]. In this method, an insulating thin layer is required in between aluminum thin wire and IGZO thin film to prevent the current flow through the sample and substrate. The aluminum wire strip act as a heater and thermometer at the same time. An alternating current (AC) was applied through the aluminum wire strip with an angular frequency of ω. Furthermore, a 10 Ω shunt resistance was connected in series to acquire the reference signal at a frequency of *ω*. The metallic wire strip was Joule heated at a frequency 2*ω*, of according to *P = I^2^R*. The metallic resistance temperature coefficient, (*1/R*) (*dR/dT*), was measured by heating the sample. With the application of an AC at *ω*, voltage oscillation at a frequency of *3ω* (denoted as *V*_3ω_) could be observed. The temperature oscillation (*T_2_*_ω_) at the frequency of 2*ω*, expressed as
(1)$$T_{2\omega }=2\frac{dT}{dR}\cdot \frac{R}{V_{1\omega }}V_{3\omega }$$

The electrical resistance is proportional to the temperature. For this reason, the temperature is also fluctuated at 2*ω* due to Joule heating at 2*ω*. The metallic wire resistance was varied with the oscillation at 2*ω*, i.e., *T*_2ω_. The thermal conductivity of the thin film, *κ*_f_, can be expressed as follows:(2)$$\kappa_{f}=\frac{Pd_{f}}{wl\cdot T_{2\omega ,f}}$$
where *P* is the AC heating power, $d_{f}$. is the thickness of the film, *w* is the width, and *l* is the length of the wire strip. A 16-bit digital-analog converter and LabVIEW^®^ system (National Instruments, Austin, TX, USA) were functioned to track the *3ω* signal. The *3ω* is ten thousand times smaller than that of the *ω* voltage signal and it is possible to measure even at an intensity. The thermal conductivity of various temperature was also measured from room temperature to 373 K.

The value of Young’s modulus was measured by the nanoindentation method (Agilent-Nano indenter DCM, Agilent, Santa Clara, CA, USA) to evaluate the average phonon group velocity in the IGZO films. The thickness of the IGZO films that were measured was 500 nm, and the indentation depth ranged from 70 to 120 nm. The density values of the films were also measured by X-ray reflectometry (XRR, Rigaku, Woodlands, TX, USA), to evaluate the average phonon velocity.

## 3. Results

The crystallinity of the IGZO thin films was verified using STEM observations. A typical result is shown in Figure 3. As shown in Figure 3a, the cross-sectional STEM images acquired from the different regions of the IGZO film show nano crystallinity. Herein, the IGZO specimen of less than 50 nm thickness was prepared by FIB. This shows the imperfect crystallinity with local variations in lattice spacing. Since such areas with non-perfect crystallinity overlap with one another along the electron beam direction, it is challenging to directly visualize the local crystallinity. Nevertheless, certain areas showed evident lattice fringes in the amorphous matrix. The selected area electron diffraction pattern in Figure 3b exhibited certain intensity maxima on the Debye ring. This feature strongly supports the formation of nanocrystals in the IGZO films. From the real and reciprocal space data, it is interpreted that (i) the average crystalline grain size was approximately 2–5 nm and (ii) the most common lattice spacing was approximately 0.24–0.26 nm (Figure 3c).

Several research groups have reported that the absorption of oxygen decreased the electrical performance of IGZO TFTs because oxygen absorption is directly associated with the oxygen vacancies in oxide thin films [31]. The resistivity is proportional to the O_2_/Ar flow ratio, and the oxygen partial pressure [PO_2_] is inversely proportional to the carrier density of films. Therefore, the carrier density in oxide thin films is linked to the oxygen vacancies in the films. The increase in the percentage of O_2_ flow rate would cause a reduction in the number of these vacancies. This would, in turn, result in a reduction in the electron density in the film and thereby, the electron thermal conductivity. However, the measured total thermal conductivity of the IGZO thin films increased with an increase in the O_2_/Ar flow ratio. So, thermal conductivity of thin films is proportional to the O_2_ flow ratio. It is indicated that the increase in the mean free path of phonons propagating in the films owing to the reduction in the number of oxygen vacancies and the consequent improvement in the nano crystallinity of the films caused the increase in thermal conductivity.

Figure 4 shows the temperature oscillation (1/R) (dR/dT) across the IGZO thin film (T2ω, f) at the frequency of 2ω. The results correspond to the temperature oscillation difference between samples and the reference. All the data show that the temperature oscillation, i.e., ∆T is constant with increasing of frequency. Figure 5 shows the thermal conductivity values of Samples A, B, and C at different temperatures (1.65, 1.76, and 2.58 W m^−1^K^−1^, at room temperature respectively). T. Yoshikawa et al. reported that the thermal conductivity of 200 nm IGZO thin film (0–2% O_2_ flow ratio) is 1.4 W m^−1^K^−1^ [32]. This result is quite similar to the thermal conductivity of sample A at room temperature. The thermal conductivity of samples B and C is higher than the reported value because of higher oxygen partial pressure of IGZO thin films. In our study, the thermal conductivity decreased with an increase in temperature. This temperature dependence is similar to that for crystalline materials because the thermal conductivity of amorphous materials increases with an increase in temperature in the room-temperature region [33,34,35,36]. Furthermore, B. Cui et al. were carried out extensive thermal conductivity analysis of IGZO thin films at low temperature and reported that the result of thermal conductivity is varying widely depending on deposition technique such as physical layer deposition, sputtering, and chemical synthesis; deposition power, deposition temperature and structural phases such as amorphous, poly-crystal, and single crystal [37]. Thin film porosity is proportional to the deposition power and inverse proportional to the thermal conductivity. Deposition temperature changes the structural phases and it is one of the important parameters, which is directly associated with phonon mean free path. The phonon mean free path of all amorphous materials is shorter than crystalline materials. Moreover, thermal conductivity is proportional to the mean free path of it.

Heat conduction in solids is possible to explain using the phonon–phonon interaction phenomenon. The thermal conductivity of phonon can be described according to the phonon gas model can be expressed using the following equation:(3)$$k=\frac{1}{3}C_{v}vl.\;$$
where k. is the thermal conductivity of phonon, $C_{v}$, v, and l are the specific heat capacity, average phonon velocity, and phonon mean free path respectively. The specific heat capacity of the thin film may be fixed and not dependent on the film thickness or chemical composition. The average phonon velocity (v) can be assumed as the sound velocity, which is calculated using the following equation:(4)$$v=\sqrt{\frac{E}{\rho_{d}}}.\;$$
where *E* is Young’s modulus and $\rho_{d}$ is the film density. Film density may vary depending on film thickness or deposition condition, but it is negligible. The thermal conductivity is directly associated with the mean free path so that it is a very important parameter. Figure 6 shows the values of Young’s modulus, which is measured by the nanoindentation method at various indentation depths. Although it is not evident why the measured values increase with the indentation depth, the average Young’s modulus of the IGZO films can be determined around 200 GPa from a narrow plateau at the indentation depth of 100 nm. The Young’s modulus and the phonon velocity were constant across the sample. The density and specific heat capacity of the IGZO films were reported to be 5.86 g.cm^−3^ and 426 Jkg^−1^K^−1^ [32,37].

The phonon mean free paths were estimated at room temperature using Equations (3) and (4), which are very narrow around the nanometer range. In Figure 5, the thermal conductivity of [PO_2_] = 65% (Sample C) was higher than those of [PO_2_] = 0% (Samples A) and [PO_2_] = 10% (Sample B). This indicates that the phonon mean free path of [PO_2_] = 65% (Sample C) was longer than others. The thermal conductivity was decreasing with an increase in temperature because the phonon–phonon scattering was increasing with an increase in temperature. This phenomenon is a common feature for all crystalline materials [38]. The mean free path had an inverse proportional relationship with the temperature. Hence, the thermal conductivity of T^–1^ dependence was negligible in amorphous materials (glass) generally significant in crystal. At low temperatures, the mean free path is intrinsically short owing to the short-range order atomic arrangement [39,40]. Therefore, the temperature dependent thermal conductivity of IGZO thin films shown in Figure 5 indicates a clear demonstration of the significant crystallinity.

## 4. Conclusions

We fabricated IGZO thin films by the DC sputtering method and investigated the mechanism of the thermal conductivity with different oxygen partial pressure from room temperature to 373 K. The thermal conductivity of IGZO thin films was widely vary based on deposition parameters and structural phase. The temperature dependence thermal conductivity results clearly indicate the common trends of typical microstructure although IGZO thin films were widely evaluated as an amorphous material. The ADF-STEM observations revealed that nanosized crystals with an average crystal size of 2–5 nm and a lattice distance of approximately 0.24–0.26 nm were formed in the amorphous matrix of the fabricated IGZO thin films. The increase in the thermal conductivity of the IGZO thin films with the increase in the O_2_/Ar flow ratios during the film deposition is interpreted to be owing to the reduction in the number of oxygen vacancies and the improved nano crystallinity of the IGZO films.

## Figures and Tables

**Figure 1 nanomaterials-11-01547-f001:**
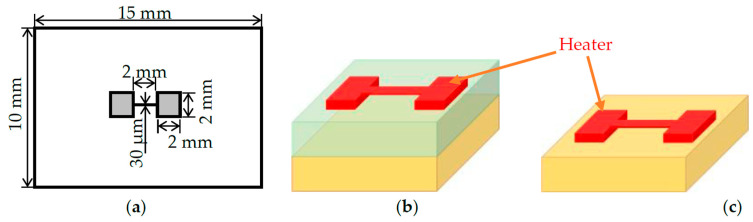
Plan view of the shadow mask of the heater with the length of 15 mm, width of 10 mm and a thickness of 20 µm (**a**) and the schematic diagram of a film sample (**b**) and the reference (**c**).

**Figure 2 nanomaterials-11-01547-f002:**
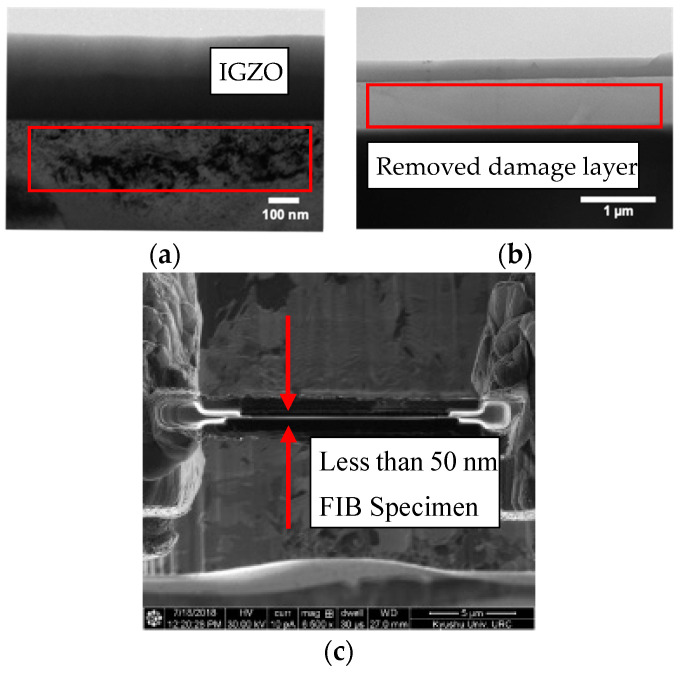
(**a**) Bright-field (BF) transmission electron microscopy (TEM) image of a cross-sectional IGZO/α-Al_2_O_3_ film indicating Ga ion beam damage (red box), (**b**) a cross-sectional BF-TEM image after subsequent low-power Ar milling that effectively removed the damaged layer on the cross-sectional specimen (red box), and (**c**) a plan-view secondary-ion microscopy image of the thin foil specimen with a thickness less than 50 nm prepared by the Ga/Ar ion beam-milled Sample B.

**Figure 3 nanomaterials-11-01547-f003:**
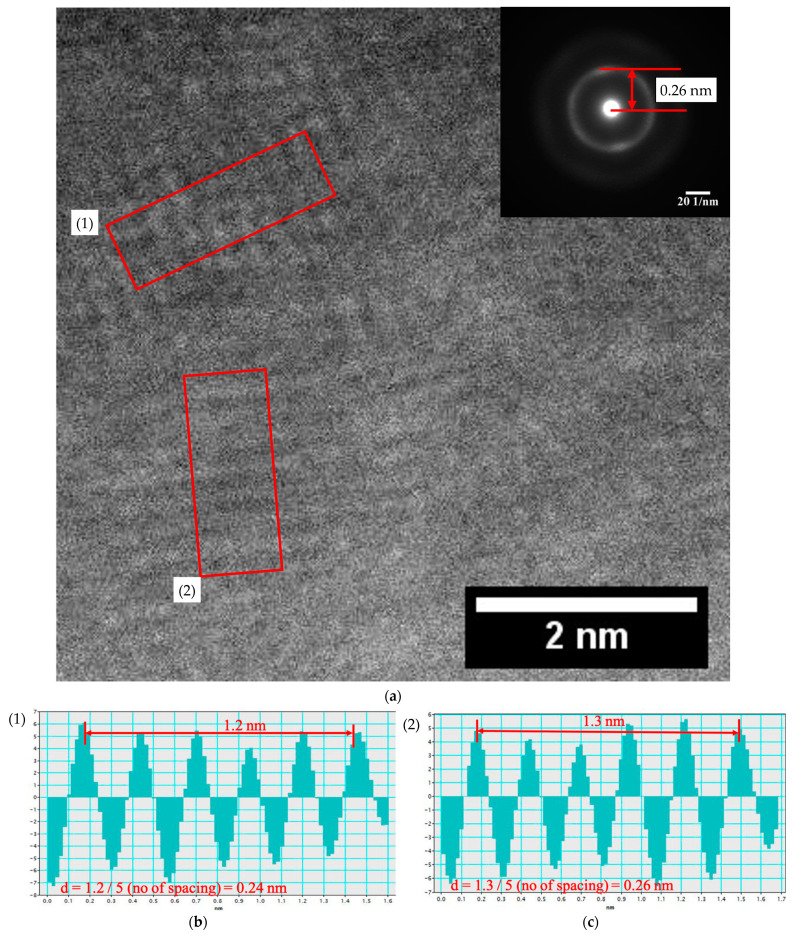
Cross-sectional scanning transmission electron microscopy (STEM) observation of Sample B. (**a**) Annular dark-field (ADF) STEM image with a selective area diffraction pattern (inset) acquired from the IGZO film under the TEM (parallel beam) mode and (**b**,**c**) lattice distance in two regions of (**a**).

**Figure 4 nanomaterials-11-01547-f004:**
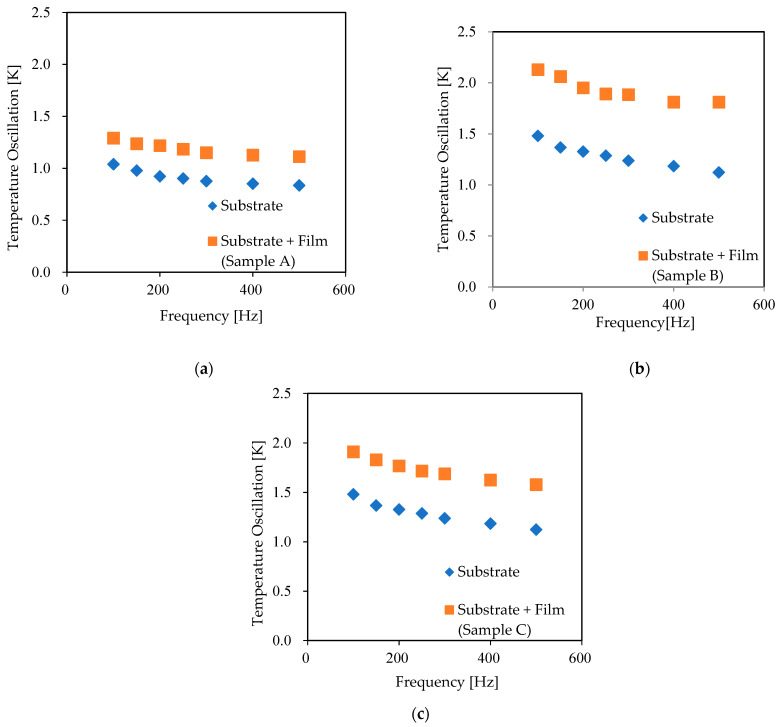
Temperature oscillation amplitude of (**a**) Sample A, (**b**) Sample B, and (**c**) Sample C at room temperature.

**Figure 5 nanomaterials-11-01547-f005:**
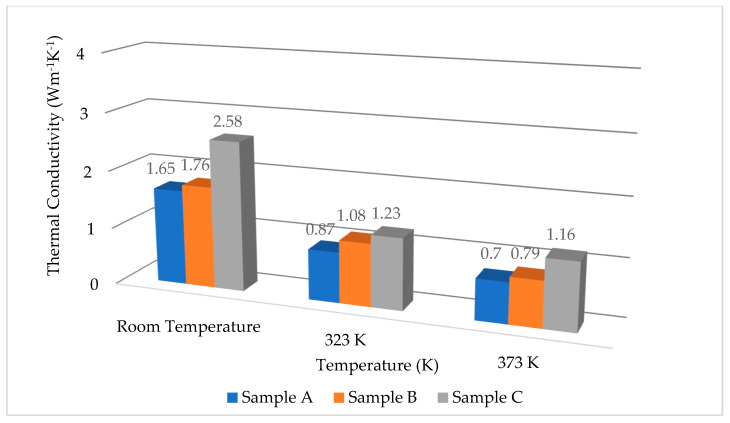
Temperature dependence of thermal conductivity of IGZO thin films in the temperature range from room temperature to 373 K.

**Figure 6 nanomaterials-11-01547-f006:**
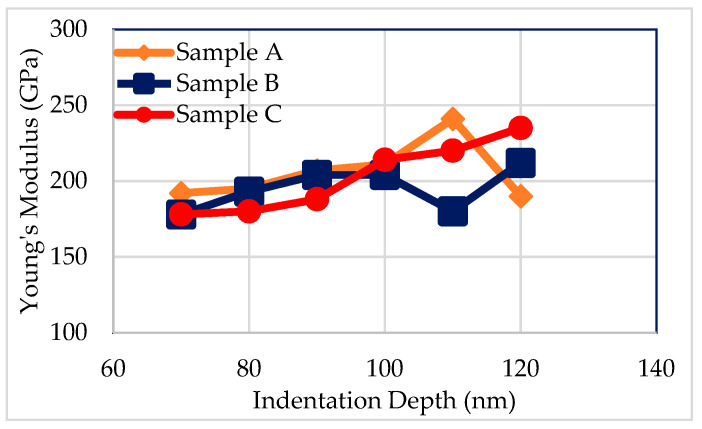
Young’s modulus with various indentation depth of 500 nm IGZO thin films.

## Data Availability

The data that support the findings of this study are available from the corresponding author upon reasonable request.

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
