# Peer review of "Thermal Conductivity of Nano-Crystallized Indium-Gallium-Zinc Oxide Thin Films Determined by Differential Three-Omega Method"

_nanomaterials, 2021, doi:10.3390/nano11061547_

Round 1
Reviewer 1 Report
This article describes temperature-dependent thermal conductivity analyzes of IGZO deposited by DC magnetron sputtering at various oxygen partial pressures PO2. It is shown that the thermal conductivities increased with PO2 and decreased with higher ambient temperatures. The authors concluded from complementary electron microscopy experiments that the nanocrystalline microstructure controls the heat conduction in IGZO films.
Although extensive thermal conductivity analyzes were carried out on IGZO ([25] B. Cui et al., “Thermal Conductivity Comparison of Indium Gallium Zinc Oxide Thin Films: Dependence on Temperature, Crystallinity, 275 and Porosity,” J. Phys. Chem. C, vol. 120, no. 14, pp. 7467–7475, 2016, doi: 10.1021/acs.jpcc.5b12105 and [26] T. Yoshikawa et al., “Thermal conductivity of amorphous indium-gallium-zinc oxide thin films,” Appl. Phys. Express, vol. 6, no. 2, 2013, 277 doi: 10.7567/APEX.6.021101) the results obtained here are not compared with those obtained in previous publications. In this article, these works are cited only in relation to the density and specific heat capacity of IGZO films. In addition, the results differ significantly from each other. In order to adhere to the principles of good scientific practice, the authors should have discussed these former thermal conductivity results and the differences.
Author Response
Dear Reviewer,
Good day,
We thank you for the time you put in reviewing our paper and we sincerely appreciate the suggestions and comments. Thank you for the positive feedback for improving our work and manuscript. The revisions have been made regarding suggestions and comments. The below are the questions extracted from the reviewer's comments along with our summarized response.
Kindly refer to the description below for the reply regarding remarks and questions.
We are trying our best to improvise our work and look forward to meeting your expectation.
Thank you very much.
Regards,
Rauf Khan
Student, Kyushu University

Reviewer 2 Report
In this paper “Thermal Conductivity of Nano-crystallized Indium–Gallium–Zinc–Oxide Thin Films Determined by Differential Three-Omega Method” Author presents investigations of nano-crystallinity Indium–Gallium–Zinc–Oxide thin films. Measurements were made using differential three-omega method. The thin films were deposited on an alumina substrate by direct current magnetron sputtering at different oxygen partial pressures (0%, 10%, and 65%). Electron microscopy observations shows presence of nanocrystals embedded in the amorphous matrix of the Indium–Gallium–Zinc–Oxide films.
In my opinion this paper is interesting and can be interesting for readers of Nanomaterials journal. The paper contains 6 figures and 4 formulas – figures are legible and good quality.
English of the paper is rather good – in my opinion the language of the paper should be a little improved. I am asking for corrections by a native speaker.
I find some mistakes for example:
- Introduction chapter – in my opinion should be correct. Authors should include new information about properties of measured materials. More information based on worldwide (global) study.
- Materials and Methods chapter – please describe equipment (simulation programs for analysis) used in the theoretical experiment – model of equipment (manufacturer, city, country).
- References chapter is sufficient but papers cited in the references 26 from all 27 are older then 5 years – these publications constitute over 96 % of all cited papers. I propose to add some new (from the last 5 years) publications on the production or properties of these type of materials. Author should include several modern papers of global research in this field.
- Minimum 22 papers from all 27 (over 81 %) are wrote by authors from Asia – mainly Japan and others. Are the topics covered in this article only by research centers from Asia? I ask for a reliable verification of global research in this field. I propose to add some new (from the last 5 years) publications on the production or properties of these type of materials. Authors should include several modern papers (also from Europe and America).
- In the list of references I found 1 papers of the Author of reviewed paper. Are these the first studies related to this subject for the Authors?
- Figure 3 – marked areas (1) and (2) are not legible – In this quality of figure the differences in the marked areas are not visible.
- Please describe all the variables found in the formulas.
- The last chapter should be named Conclusion not Discussion – please correct
And editions errors:
- In the whole paper, you write “0%” (for example in the line 19) – you should write this unit with a space as “0 %”.
- You write “5.86 g cm-3” (for example in the line 180) – you should write this unit with a space as “5.86 g‧cm-3”.
The results obtained are interesting and promising. The manuscript can be accepted for publication in Nanomaterials journal after MINOR corrections.
Author Response
Dear Reviewer,
Good day,
We thank you for the time you put in reviewing our paper and we sincerely appreciate the suggestions and comments. Thank you for the positive feedback for improving our work and manuscript. The revisions have been made regarding suggestions and comments. The below are the questions extracted from the reviewer's comments along with our summarized response.
Kindly refer to the attached file for the reply regarding remarks and questions.
We are trying our best to improvise our work and look forward to meeting your expectation.
Thank you very much.

Round 2
Reviewer 1 Report
Since the authors did not follow the “Comments and Suggestions for Authors
” of the reviewer, the paper should be rejected. The differences in the production of IGZO films listed in the authors response letter still do not explain why the thermal conductivities of IGZO in their investigations differ significantly from previous observations. For a reader there is still a lack of explanations why the thermal conductivities in IGZO can vary widely. Even if there are significant differences in technology in the production of IGZO films, it should be possible to compare material properties. At least the authors could have listed parts of their discussions in the reply letter in the revised paper.
In addition, it still has to be noted, in order to adhere to the principles of good scientific practice, the authors should have discussed these former thermal conductivity results and the differences.
Authors' Response:
In this study, the temperature dependence of the thermal conductivity of IGZO thin films was measured using the differential three-omega (3ω) method which reveals a clear conclusion on the crystalline phase of IGZO thin films. To adhere to the principles of good scientific practice, we tried to discuss the significant difference of our experimental results from the previous observation. I have added some important information to clarify the reason for the variation of thermal conductivity of IGZO thin film widely. This explanation will help the readers to understand the thermal transport mechanism and the phase structure of IGZO thin film elaborately.
Round 3
Reviewer 1 Report
I believe the manuscript has been sufficiently improved to warrant publication in Nanomaterials.